# Bio-Fabrication of Bio-Inspired Silica Nanomaterials from Orange Peels in Combating Oxidative Stress

**DOI:** 10.3390/nano12183236

**Published:** 2022-09-18

**Authors:** Mosleh Mohammad Abomughaid

**Affiliations:** Medical Laboratory Sciences Department, College of Applied Medical Sciences, University of Bisha, Bisha 67714, Saudi Arabia; moslehali@ub.edu.sa or mabomughaid@gmail.com

**Keywords:** silica nanoparticles, characterization, orange peel, antioxidant activity

## Abstract

Silica nanoparticles were synthesized using the aqueous extract of orange peels by the green chemistry approach and simple method. The physicochemical properties such as optical and chemical banding of as-synthesized silica nanoparticles were analyzed with UV–visible spectroscopy and Fourier transform infrared spectroscopy. Scanning Electron Microscopy with Energy Dispersive X-Ray Analysis and X-ray diffraction analysis were employed to confirm the shape, size and elemental purities of the silica nanoparticles. The thermal stability and mass loss of the silica nanoparticles was examined using thermogravimetric analysis and zeta potential analysis. The surface plasmon resonance band of the silica nanoparticle was obtained in the wavelength of 292 nm. Silica nanoparticles with a spherical and amorphous nature and an average size of 20 nm were produced and confirmed by X-ray diffraction and Scanning Electron Microscopy. The zeta potential of the silica nanoparticles was −25.00 mV. The strong and broad bands were located at 457, 642 and 796 cm^−1^ in the Fourier transform infrared spectra of the silica nanoparticles, associated with the Si–O bond. All the results of the present investigation confirmed and proved that the green synthesized silica nanoparticles were highly stable, pure and spherical in nature. In addition, the antioxidant activity of the green synthesized orange peel extract mediated by the silica nanoparticles was investigated with a DPPH assay. The antioxidant assay revealed that the synthesized silica nanoparticles had good antioxidant activity. In the future, green synthesized silica nanoparticles may be used for the production of nano-medicine.

## 1. Introduction

Nanotechnology is an innovative and developing technology with huge applications. It includes the synthesis and application of nanomaterials having sizes in the range of 1 to 100 nm [1]. In general, nanomaterials are achieved through physical and chemical methods and these methods required huge energy inputs and toxic chemicals [2]. The word green synthesis refers the production of nanomaterials using the natural substances or extract of plants or their metabolites. The metabolites from natural strains or phytochemicals from plants (alkaloids, flavonoids, terpenoids, aldehydes and amides) act as a capping/stabilizing/reducing agent. Nanomaterials biosynthesized through the green chemistry approach are less toxic, eco-friendly, reliable, sustainable and significant for pharmaceutical and other applications [3]. The green synthesis of nanoparticles via the nanobiotechnology approach has an important role in boosting production compared to chemical and physical metho [4]. There are different types of nanomaterials, namely, Cu, Zn, Au, Ag, Si, Pt, Fe, Te, Mg, Au–Ag alloy and quantum dots produced through the biological method. The green synthesized nanomaterials show biological properties such as antimicrobial activities, anticancer activities, antioxidants, etc. [5].

Silica is a complex and abundant family of materials and presents as a component of various minerals/synthetic materials such as silica-gel, aerogels, fused-quartz and fumed silica [6]. In recent years, scientists and researchers have very much been interested in doing research on silica nanomaterials due to their distinctive and versatile physiochemical character [7]. Silica nanomaterials are mainly used in different areas because of their large specific surface area, good biocompatibility, controlled particle size, pore volume, hydrophilic nature and inexpensive large-scale production method [8,9]. Moreover, silica nanoparticles have been used for disease diagnosis and treatments [10].

Silver nanoparticles have been produced and characterized using the extract of orange peels via the microwave-assisted green synthesis method [11]. Castro et al. [12] roduced silver and platinum nanomaterials by employing orange peel extract. Orange-peel-mediated Fe_2_O_3_ nanoparticles were prepared and used for cadmium removal from wastewater [13]. TiO_2_ nanoparticles were produced via the green synthesis method using orange peel extract and were used for humidity sensor applications [14]. Orange peel extract-mediated CeO_2_ nanoparticles were synthesized by [15], who evaluated their antioxidant activity and photocatalytic activity. The extract of orange peels acts as a capping and reducing agent for the synthesis of nanoparticles because it contains citric acid [16]. Orange peel extract is used in the preparation of mosquito repellent, bath oil, face creams and room-freshening sprays. The biomass of orange peels has a huge amount of natural antioxidant bioactive compounds such as resins, phenols, saponins, monoterpenes and flavonoids, which makes them suitable for medicinal and chemical purposes [17].

An orange is a citrus fruit and has good natural antioxidant properties due to the presence of alkaloids, phenols, flavonoids, etc. The peel of an orange has rich phytochemicals which are responsible for the bio-reductants involved in the synthesis of silica nanoparticles. The efficient reduction and stabilization of the phytochemicals occurred due to the presence of citric acid and ascorbic acid in the citrus, which are involved in the formation of nanomaterials. Hence, the orange-peel-mediated nanoparticles have good antioxidant activity, and they would be most significant in medicinal field.

In this work, we produced biogenic silica oxide nanomaterials using the aqueous extract of orange peels via the green synthesis method. Moreover, the physicochemical properties of the as-synthesized silica nanoparticles were determined using various analytical techniques. In addition, the antioxidant properties of the orange-peel-mediated silica nanoparticles were investigated using the DPPH assay.

## 2. Experiments

### 2.1. Collection of Orange Peel and Chemicals

A fresh orange peel was collected from the fruit market. All chemicals and reagents (99.9% of purity) were analytical grade and were employed without any purification. Milli-Q water was used throughout the investigation. Tetraethyl orthosilicate, Ethanol and Hydrochloric acid were purchased for this investigation.

### 2.2. Preparation of Orange Peel Extract

A total of 10 g of the orange peel and Milli-Q water were used for the preparation of the extract. The orange peel was washed with tap and distilled water, respectively. The samples were grinded using mortar and pestle using 250 mL of Milli-Q water. The extract was heated using a water bath at 80 °C for 30 min. After that, the extract was purified by centrifugation, then stored in a sterile container for further investigation.

### 2.3. Biogenic Synthesis of Silica Nanoparticles

The aqueous extract of the orange peel (250 mL) was added with 20 mL of Tetraethyl orthosilicate as a precursor. The mixture was stirred continuously for 15 min at room temperature. A total of 10 mL of ethanol were added to the mixture under stirring conditions for 15 min. Next, 12.5 mL of HCl were added to the mixture under stirring conditions. Finally, a precipitation with a jelly-like form was formed. The precipitation was collected and was left to dry at 90 °C for 18 h. At last, the white-color powder was attained and stored in sterile glass for further investigation. The extract of the orange peel was used as a reducing/stabilizing/capping agent for the formation of the biogenic silica nanoparticles. A schematic of the synthesis of silica nanoparticles using the extract of the orange peel is shown in Figure 1.

### 2.4. Analysis of Optical Property and Functional Groups

The optical properties of the silica nanoparticles and extract of the orange peel were assessed using a single-beam UV–vis spectrophotometer. Absorption spectra of the orange-peel-mediated silica nanoparticles were taken between 200 and 800 nm, using a single-beam C10082MD (Hamamatsu, Japan) UV–Visible spectrophotometer. The Milli-Q water was used as the reference. A Fourier transform infrared spectroscopy (FT-IR) analysis in the attenuated total reflectance (ATR) mode was performed using a Frontier FT-IR spectrophotometer (Bruker optics, Woodlands, TX, USA) to confirm the contribution of the constituents from the extract of the orange peel in reducing and fabricating the silica nanoparticles. UV–Vis and FT-IR spectra were plotted with the help of Origin software (Origin 9.8, (2021) OriginLab Corporation, Northampton, MA, USA).

### 2.5. Analysis of Shape, Elemental Composition and Nature of Silica Nanoparticles

The shape of the silica nanoparticles was studied using Scanning Electron Microscopy (JEOL, Tokyo, Japan). The elemental composition of the biogenic silica nanoparticles was analyzed using the Epsilon Xflow (Malvern P Analytical, Ltd., Malvern, UK) energy-dispersive X–ray analyzer (EDAX). The nature of the silica nanoparticles was studied using the X-ray diffraction technique using Cu Ká radiation.

### 2.6. Zeta Potential and Thermal Stability Analysis

The dynamic light scattering technique was used for the zeta potential analysis of the silica nanoparticles by the Zetasizer ultra, (Malvern Panalytical, Malvern, UK) size range from 0.1 nm to 10,000 nm). The thermal decomposition of the synthesized silica nanoparticles was characterized by a thermogravimetric analysis (TGA) (Photometrics, Inc. Huntington Beach, CA, USA) to verify the changes or weight loss that the silica nanoparticles underwent during their heating at various temperatures.

### 2.7. Antioxidant Activity

The antioxidant activity of the orange-peel-mediated silica nanoparticles was investigated using the 2,2-diphenyl-1-picrylhydrazyl (DPPH) method. Various concentrations of the silica nanoparticles were formulated for this assay. A total of 0.1 mM of (700 µL) DPPH reagents were mixed with 100 µL of the tested solution and the solution was preserved from the light. Methanol and water were used as the standard solution. After half an hour of incubation, the absorbance value was measured at wavelength of 515 nm. The ability to reduce free DPPH radicals was measured according to Nguyen et al. [18].

## 3. Results and Discussion

### 3.1. Characterization of Silica Nanoparticles

The optical property of the prepared orange-peel-mediated silica nanoparticles was determined using a UV–visible spectrophotometer under the region of 200–800 nm, which acquired an absorption peak λ max at 292 nm, as shown in Figure 2. Verma and Bhattacharya [19] produced silica nanoparticles using Tetraethyl orthosilicate via the hydrolysis method and obtained an absorption peak λ max at 270 nm in their UV–visible spectroscopic analysis.

The FTIR measurement of the silica nanoparticles was conducted to discover the interaction between the phytomolecules of the extract and the nanoparticles. The analysis of the FT-IR illustrated the binding properties of the phytoconstituents responsible for stabilizing and capping the nanoparticles. The major peaks at 947 and 1048 cm^−1^ were ascribed to Si–O–Si bonds. The strong peaks at 457, 642 and 796 were attributed to the presence of symmetric Si–O bonds and the stretch of Si–O in the green synthesized silica nanoparticles. Other peaks such as 1643 and 3419 cm^−1^ indicated the occurrence of hydroxyl and carbonyl groups (Figure 3). An FT-IR study on orange peel exact was performed by Niluxsshun et al. [20]. They reported the stretching frequency of the O–H functional group, the C=O stretching of the carbonyl group, aromatic C=C stretching vibration and that an alkyne group was present in the phytochemicals of the orange peel exact. In addition, an FT-IR study on free silica nanoparticles was performed by Stanley and Nesaraj [21] and they determined the presence of an Si–O bending vibration and the Si–O–Si plane, which was compared to the orange peel exact mediated by silica nanoparticles. The result was consistent with the earlier investigation [22]. They synthesized silica nanoparticles via green chemistry using the extract of the *Thuja orientalis* leaf and confirmed the formation of a Si–O stretch via FT-IR analysis.

The XRD image of the silica nanoparticle is shown in Figure 4. The XRD pattern shows the occurrence of an intense peak at 2θ = 22°, which confirms the formation of amorphous silica nanoparticles. No diffraction peaks/impurities peaks were observed in the XRD pattern of the prepared silica nanoparticles. The result of the XRD pattern matched with the JCPDS file for silica nanoparticles and it determined the pure form of the silica nanoparticles. The average size of the nanoparticles was found to be 20 nm. This result is similar to Ghani et al. [23], who performed an analysis of XRD and proved the formation of silica nanoparticles with an amorphous nature.

The size and morphology of the silica nanoparticles were determined using SEM with different magnification scales. Typical SEM images of the silica nanoparticles fabricated with the extract of an orange peel are displayed in Figure 5a,b. The particles were mostly spherical in shape and few particles were agglomerated. The average sizes were measured and most of the particles were nearly 18–20 nm in size. However, Piela et al. [24] described the biogenic synthesis of silica nanoparticles using corn cob husks and found that the silica particles were spherical in the variety of 40–70 nm. The chemical composition and purity of green synthesized silica nanoparticles were assessed using an EDX analysis (Figure 6). The existence of intense peaks of Si (62%) and O (38%) were observed in the EDX spectrum, which confirmed the formation of silica nanoparticles.

The stability of the nanoparticles was measured using a zeta potential analysis. The zeta potential value of the synthesized silica nanoparticles was predicted to be −25.0 mV (Figure 7). The negative charge indicates the good stability of the nanoparticles and indicates less agglomeration. This result is consistent with the silica nanoparticle preparation from *Cynodon dactylon*, which had nanoparticles with a −30.3 mV zeta potential value [25] The reduction and capping action of the phytomolecules that occurred in the plant extract might be a reason for the negative zeta potential value of the silica nanoparticles.

The thermal decomposition and stability of the powder of silica nanoparticles were investigated via thermogravimetric analysis. Figure 8 depicts the thermogravimetric analysis and thermal differential analysis of the silica nanoparticles. A TGA determined the two stages of weight loss. Nearly 30.5% of the weight loss occurred between 250 °C and 950 °C because of the degradation of the organic matter derived from the plant extract. The Differential Thermal Analysis (DTA) curve of the silica nanoparticles illustrates the exothermic peaks, which are attributed to the loss of mass at different temperatures, and proves the degradation of the hydroxyl groups and organic matter. These peaks clearly reveal the loss of mass in the as-synthesized silica nanoparticles at various temperatures. In addition, they prove that organic matter from the plant extract was involved in the formation of the silica nanoparticles and that it was present on the surface of the synthesized silica nanoparticles. Maroušek et al. [26] reported the degradation of organic matter, organic solvents and hydroxyl groups on lignin-mediated silica nanoparticles using the TGA method.

### 3.2. Antioxidant Activity

Orange-peel-mediated silica nanoparticles were used to assess their antioxidant activity via a DPPH assay (Table 1). The low concentration (50 µg/mL) of the silica nanoparticles exhibited a least percentage (22.5%) of free radical scavenging activity, whereas the maximum concentration (500 µg/mL) of the silica nanoparticles showed the extreme percentage (90.72%) of the free radical inhibition. The maximum percentage of the free radical inhibition was observed due to the presence of bioactive compounds on the surface of the nanoparticles. Dong et al. [27] investigated the physico-chemical properties and antioxidant activity of chitosan film fused with modified silica nanoparticles as an active food packaging.

## 4. Conclusions

Silica nanoparticles were synthesized with the help of a green chemistry approach. The aqueous extract of an orange peel acted as a stabilizing and capping agent for the production of the silica nanoparticles. Orange-peel-mediated silica nanoparticles have excellent physicochemical properties, as was confirmed by different techniques. This approach promotes the utilization of biological resources for the production of silica nanomaterials through eco-friendly and less toxic methods. The current nano-biotechnological method is capable of fabricating silica nanoparticles. Moreover, this investigation effectively proves the convenient utilization of orange peel extract as a template to obtain stable silica nanoparticles. The antioxidant activity of orange-peel-mediated silica nanoparticles was demonstrated using a DPPH assay, which confirmed that the synthesized nanoparticles have good antioxidant properties.

## Figures and Tables

**Figure 1 nanomaterials-12-03236-f001:**
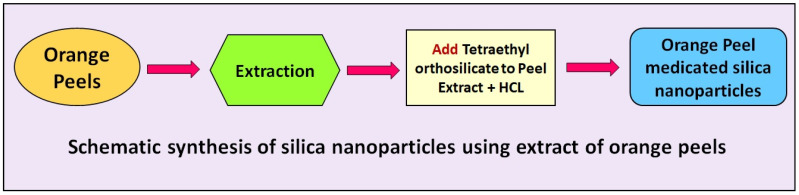
Schematic synthesis of silica nanoparticles using extract of orange peel.

**Figure 2 nanomaterials-12-03236-f002:**
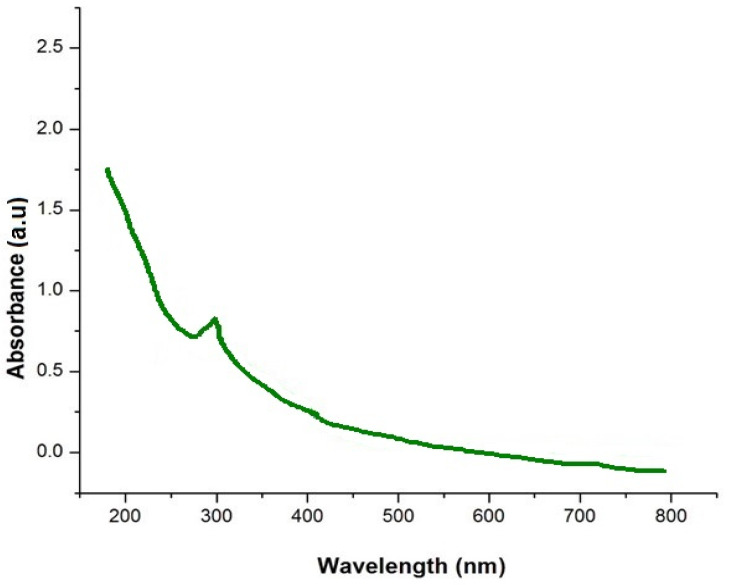
UV spectra analysis of orange-peel-mediated silica nanoparticles.

**Figure 3 nanomaterials-12-03236-f003:**
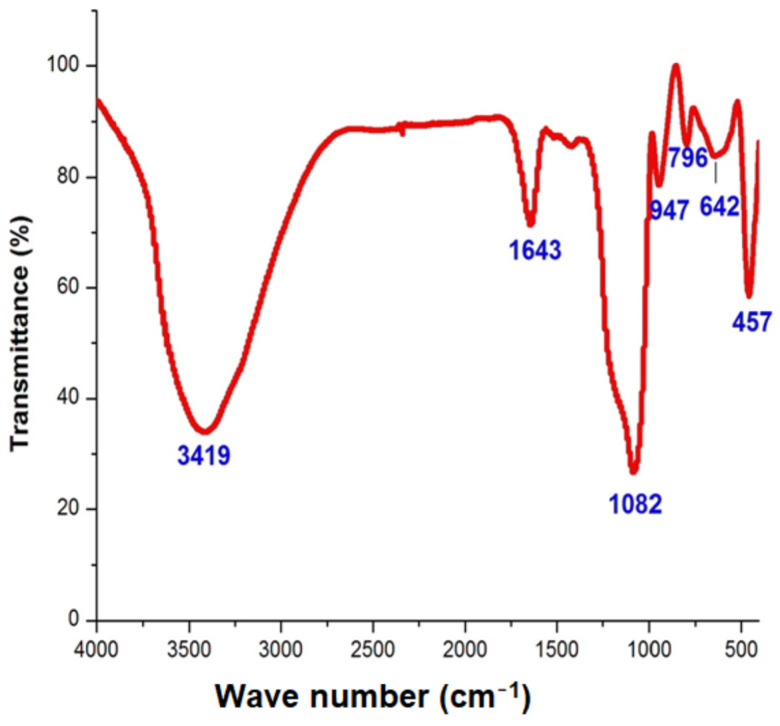
FT-IR analysis of orange-peel-mediated silica nanoparticles.

**Figure 4 nanomaterials-12-03236-f004:**
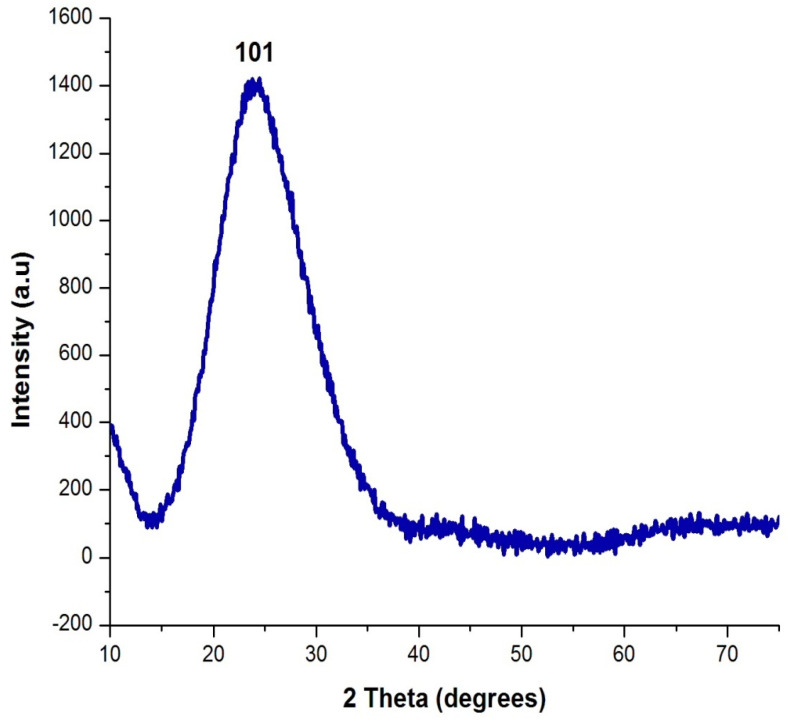
XRD analysis of orange-peel-mediated silica nanoparticles.

**Figure 5 nanomaterials-12-03236-f005:**
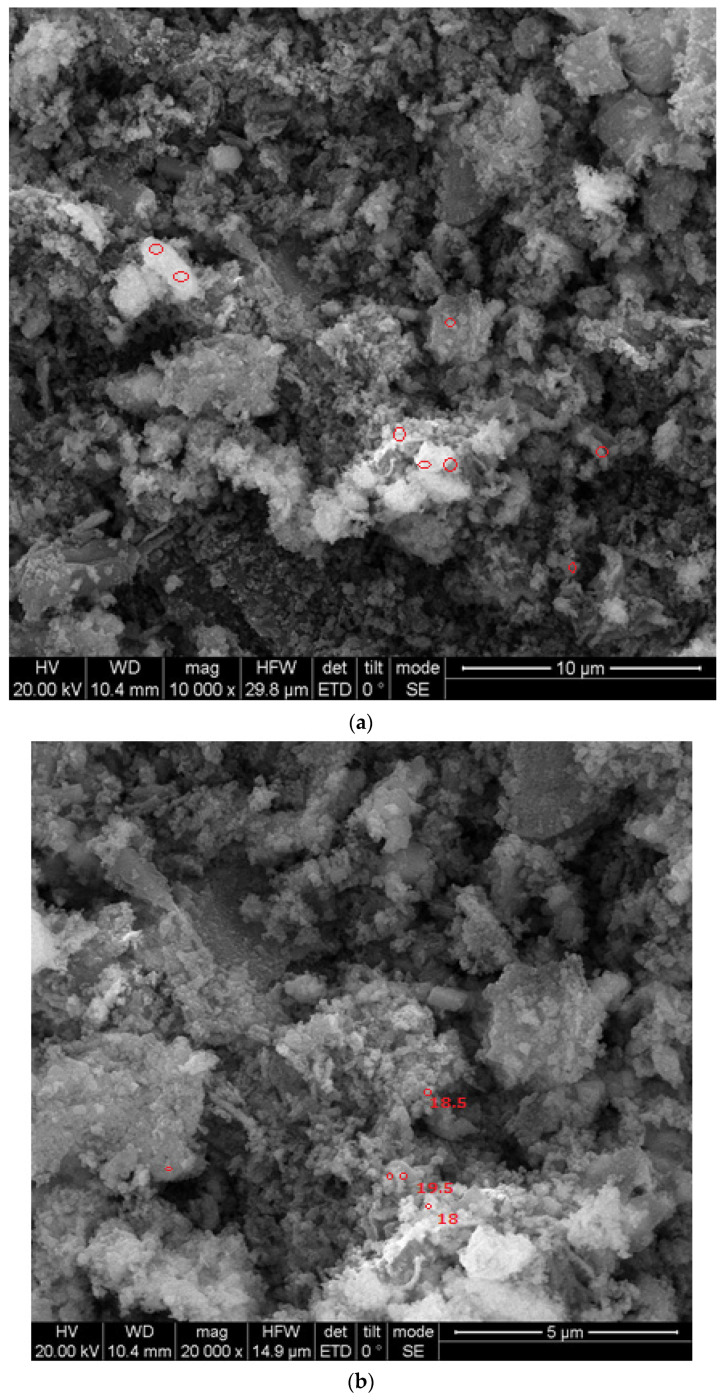
(**a**,**b**) SEM analysis of orange-peel-mediated silica nanoparticles. The SEM revealed the monodispersed distribution of particle sizes in the surface morphology, as well as the size of orange peel silica nanoparticles.

**Figure 6 nanomaterials-12-03236-f006:**
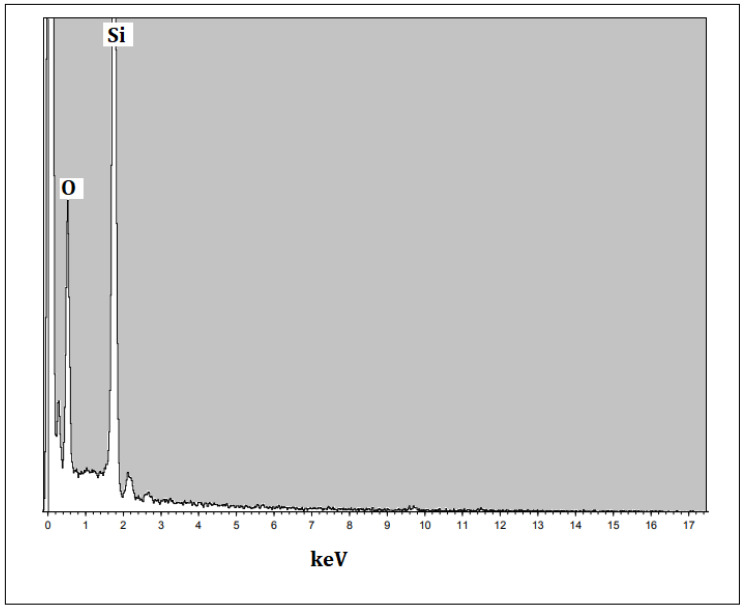
EDX analysis of orange-peel-mediated silica nanoparticles.

**Figure 7 nanomaterials-12-03236-f007:**
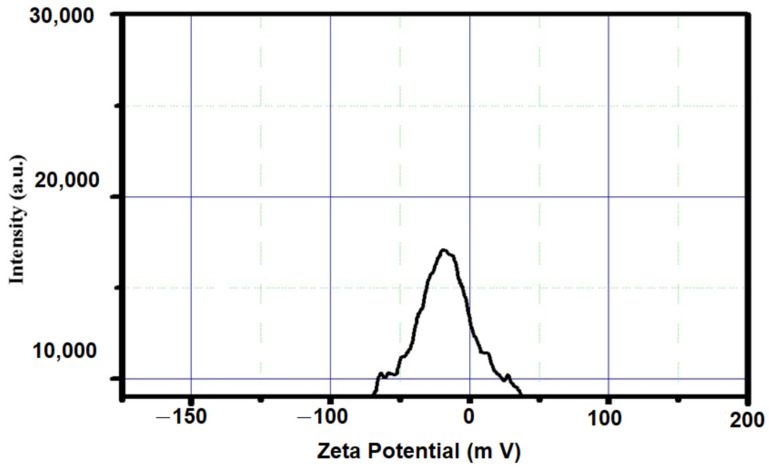
Zeta potential analysis of orange-peel-mediated silica nanoparticles.

**Figure 8 nanomaterials-12-03236-f008:**
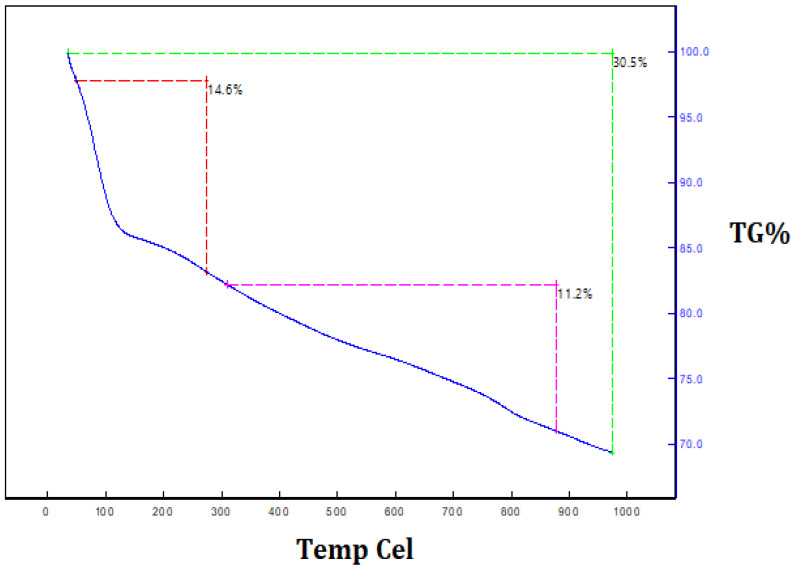
Thermogravimetric analysis of orange-peel-mediated silica nanoparticles.

**Table 1 nanomaterials-12-03236-t001:** Antioxidant activity of orange-peel-mediated silica nanoparticles.

S.No.	Concentration of Orange-Peel-Mediated Silica Nanoparticles (µg/mL)	Free Radical Scavenging Activity (%)
1	50	22.5 ± 0.12
2	100	36.7 ± 0.28
3	200	65.35 ± 0.32
4	500	90.72 ± 0.45

## Data Availability

Data available on request due to restrictions e.g., privacy or ethical.

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
