# Peer review of "Bio-Fabrication of Bio-Inspired Silica Nanomaterials from Orange Peels in Combating Oxidative Stress"

_nanomaterials, 2022, doi:10.3390/nano12183236_

Round 1
Reviewer 1 Report
The study, titled “Bio‑Fabrication of Bio‑Inspired Silica Nanomaterials from Orange Peel in Combating Oxidative Stress” discusses the green chemistry based on simple methods used to develop silica nanoparticles from orange peel aqueous extract and is well characterized by UV, SEM, XRD and TGA. Further, the orange peel antioxidant properties using a DPPH assay, extract-mediated silica nanoparticles were examined. Therefore, a recommendation for publishing this paper in nanomaterials after the major revision.
Comments:
1. The orange peel-mediated nanoparticles were reported by several researchers, as the authors noted in their introduction. Utilizing orange peel extract, silica nanoparticles were synthesized in this study. How to overcome limitations such as solubility, stability, and antioxidant activity?
2. An introduction section will include a discussion of the significance of orange peel nanoparticles.
3. The silica nanoparticles mediated by orange peel are used to have a limited characterization. Consequently, a confirmation study of orange peel mediated silica nanoparticles utilizing free orange peel exact, free silica nanoparticles, and orange peel mediated silica nanoparticles was conducted as suggested.
4. The morphology of free silica nanoparticles, free orange peel exact, and orange peel mediated silica nanoparticles are compare using SEM analysis. Additionally, morphological changes can be more precisely identified with TEM investigation.
5. DLS data required additional data for the size of orange peel mediated silica nanoparticles.
6. To confirm that orange feels exact is mediated by silica nanoparticles by evaluating the FT-IR study for orange feels exact and free silica nanoparticles.
7. A recommended illustrative diagram of orange peel-mediated silica nanoparticles for antioxidant activity and a schematic synthesis route of particles should be included in the manuscript. It would improve the quality of the manuscript.
Author Response
Author response to reviewer’s comments
The study, titled “Bio‑Fabrication of Bio‑Inspired Silica Nanomaterials from Orange Peel in Combating Oxidative Stress” discusses the green chemistry based on simple methods used to develop silica nanoparticles from orange peel aqueous extract and is well characterized by UV, SEM, XRD and TGA. Further, the orange peel antioxidant properties using a DPPH assay, extract-mediated silica nanoparticles were examined. Therefore, a recommendation for publishing this paper in nanomaterials after the major revision.
Dear Reviewer, thank you for your valuable comments to improve the quality of the manuscript.
Comments:
- The orange peel-mediated nanoparticles were reported by several researchers, as the authors noted in their introduction. Utilizing orange peel extract, silica nanoparticles were synthesized in this study. How to overcome limitations such as solubility, stability, and antioxidant activity?
Dear Reviewer, Thank you for the suggestion. I have faced a few issues on the solubility of orange peel-mediated nanoparticles. Then, I followed the standard protocol for sonication to prepare the nanosuspension for antioxidant studies. I didn’t face any other issues in stability.
- An introduction section will include a discussion of the significance of orange peel nanoparticles.
Thank you for the suggestion. The discussion of the significance of orange peel nanoparticles has been included in the Introduction section.
.3. The silica nanoparticles mediated by orange peel are used to have a limited characterization. Consequently, a confirmation study of orange peel mediated silica nanoparticles utilizing free orange peel exact, free silica nanoparticles, and orange peel mediated silica nanoparticles was conducted as suggested.
Dear Reviewer, I hearty thank you for your suggestion. There are so many reports available and published about characterization for free silica nanoparticles. Hence I didn’t perform for Free silica nanoparticles.
- The morphology of free silica nanoparticles, free orange peel exacts, and orange peel mediated silica nanoparticles are compare using SEM analysis. Additionally, morphological changes can be more precisely identified with TEM investigation.
Dear Reviewer, I agree your comparison study of SEM analysis for free silica nanoparticles, free orange peel exact, and orange peel mediated silica nanoparticles. But, the free silica nanoparticles were spherical, which were obtained by chemical method and orange peel mediated silica nanoparticles were spherical in nature. The morphology of free orange peel exact could not investigated because it is paste like appearance. TEM investigation will be carried out in future. Thank you for your valuable suggestion.
- DLS data required additional data for the size of orange peel mediated silica nanoparticles.
Dear Reviewer, Thank you for your suggestion. Sorry, I didn’t perform the DLS study. Here no facility in my research institute. I will do it for my future studies.
- To confirm that orange peels exact is mediated by silica nanoparticles by evaluating the FT-IR study for orange peels exact and free silica nanoparticles.
Dear Reviewer, Thank you for your suggestion. I have not done the FT-IR study for orange peels exact and free silica nanoparticles. But your suggestion is valuable. Hence in our previous study I compare it. I have included the comparison report in the manuscript. Kindly consider it. Thank you
FT-IR study for orange peels exact has been performed by Niluxsshun et al., (2021). They reported the functional group of O-H stretching frequency, C=O stretching of the carbonyl group, aromatic C=C stretching vibration and alkyne group present in phytochemicals of the orange peels exact. Hence, I compared with above result. I thank you for your valuable suggestion.
In addition, FT-IR study for free silica nanoparticles was done by Stanley and Nesaraj (2014) and they determine the presence of the Si-O bending vibration and Si-O-Si plane, which was compared for orange peels exact mediated by silica nanoparticles.
Stanley, R. and Nesaraj, A.S., 2014. Effect of surfactants on the wet chemical synthesis of silica nanoparticles. International Journal of Applied Science and Engineering, 12(1), pp.9-21.
Niluxsshun, M.C.D., Masilamani, K. and Mathiventhan, U., 2021. Green synthesis of silver nanoparticles from the extracts of fruit peel of Citrus tangerina, Citrus sinensis, and Citrus limon for antibacterial activities. Bioinorganic chemistry and applications, 2021.
- A recommended illustrative diagram of orange peel-mediated silica nanoparticles for antioxidant activity and a schematic synthesis route of particles should be included in the manuscript. It would improve the quality of the manuscript.
The illustrative diagram and schematic synthesis have been included in the manuscript as per your suggestion. (Figure has been included in the manuscript).
Reviewer 2 Report
In the manuscript, the authors presented a green chemistry approach for synthesis of silica nanoparticles. In this approach, the aqueous extract of orange peel was used as reducing/stabilizing/capping agent for the production of silica nanoparticles. Moreover, the antioxidant activity of orange peel mediated silica nanoparticles was conducted using DPPH assay and confirmed that synthesized nanoparticles have good antioxidant properties. The manuscript is interesting, and important, and thus I recommend to accept the manuscript after some revisions.
1. In the manuscript, the aqueous extract of orange peel was used as reducing agent for the production of silica nanoparticles. We want to know the specific compounds in the aqueous extract of orange peel were used as reducing agent.
2. In Figure 4a and 4b, the authors stated that the average sizes of the particles were nearly 18-20 nm in size. However, we cannot confirm the size by the SEM images in Figure 4a and 4b.
3. The silica nanoparticles synthesized in this manuscript have been applied as antioxidant agent. Is the antioxidant activity originated from the compounds in extract of orange peel? The antioxidant activity of silica nanoparticles synthesized by other approach should be investigated.
Author Response
Comments and Suggestions for Authors
In the manuscript, the authors presented a green chemistry approach for synthesis of silica nanoparticles. In this approach, the aqueous extract of orange peel was used as reducing/stabilizing/capping agent for the production of silica nanoparticles. Moreover, the antioxidant activity of orange peel mediated silica nanoparticles was conducted using DPPH assay and confirmed that synthesized nanoparticles have good antioxidant properties. The manuscript is interesting, and important, and thus I recommend to accept the manuscript after some revisions.
Dear Reviewer, thank you for your valuable comments to improve the quality of the manuscript.
- In the manuscript, the aqueous extract of orange peel was used as reducing agent for the production of silica nanoparticles. We want to know the specific compounds in the aqueous extract of orange peel were used as reducing agent.
Thank you for your suggestion. I have produce the nanoparticles using the crude aqueous extract of orange peel. It has mixture of compounds like alkaloids, phenols and flavonoids. Sorry, I could not identify the specific compounds in the mixture. In the future of my study, I will have focused according to your advice.
- In Figure 4a and 4b, the authors stated that the average sizes of the particles were nearly 18-20 nm in size. However, we cannot confirm the size by the SEM images in Figure 4a and 4b.
Thank you for your suggestion. Size has been marked in SEM images in Figure 4a and 4b.
- The silica nanoparticles synthesized in this manuscript have been applied as antioxidant agent. Is the antioxidant activity originated from the compounds in extract of orange peel? The antioxidant activity of silica nanoparticles synthesized by other approach should be investigated.
Yes, the antioxidant activity is originated from the compounds in extract of orange peel. The other approach will be carried out in my future investigation of anticancer activity. Thank you for your suggestion.
